# Electrochemical Soil Nitrate Sensor for In Situ Real-Time Monitoring

**DOI:** 10.3390/mi14071314

**Published:** 2023-06-27

**Authors:** Mohammed A. Eldeeb, Vikram Narayanan Dhamu, Anirban Paul, Sriram Muthukumar, Shalini Prasad

**Affiliations:** 1Department of Bioengineering, University of Texas at Dallas, Richardson, TX 75080, USA; 2EnLiSense LLC, Allen, TX 75013, USA; sriramm@enlisense.com

**Keywords:** electrochemical sensing, in situ soil nitrate sensor, real-time continuous soil monitoring, soil texture triangle

## Abstract

Sustainable agriculture is the answer to the rapid rise in food demand which is straining our soil, leading to desertification, food insecurity, and ecosystem imbalance. Sustainable agriculture revolves around having real-time soil health information to allow farmers to make the correct decisions. We present an ion-selective electrode (ISE) electrochemical soil nitrate sensor that utilizes electrochemical impedance spectroscopy (EIS) for direct real-time continuous soil nitrate measurement without any soil pretreatment. The sensor functionality, performance, and in-soil dynamics have been reported. The ion-selective electrode (ISE) is applied by drop casting onto the working electrode. The study was conducted on three different soil textures (clay, sandy loam, and loamy clay) to cover the range of the soil texture triangle. The non-linear regression models showed a nitrate-dependent response with R^2^ > 0.97 for the various soil textures in the nitrate range of 5–512 ppm. The validation of the sensor showed an error rate of less than 20% between the measured nitrate and reference nitrate for multiple different soil textures, including ones that were not used in the calibration of the sensor. A 7-day-long in situ soil study showed the capability of the sensor to measure soil nitrate in a temporally dynamic manner with an error rate of less than 20%.

## 1. Introduction

The rapid rise in food demand is straining our soil to the point that the United Nations (UN) has food insecurity, desertification, land degradation, and ecosystem imbalance as some of the critical problems to conquer in its Sustainable Development Goal (SDG) report [1]. A major solution to these problems is sustainable agriculture. The National Institute of Food and Agriculture (NIFA) defines sustainable agriculture as a system that integrates the production of plants and animals that over the long term would satisfy human food requirements, protects the environment and enhances its natural resources, efficiently uses nonrenewable resources complimented by the natural biological cycle, and improves the quality of life for society and farmers [2]. However, to achieve this goal, farmers need to know the health of their soil throughout the year [3,4,5]. The current standard for measuring soil nitrate is the use of a cadmium reduction method which is costly and requires days or weeks to transfer the sample to the lab and then pretreat it and analyze it. Due to the high expenses of soil analysis, farmers usually test their land once every two to five years. This has led to a surge in research on cheaper in situ alternatives to current lab analysis techniques for continuous measurement of soil nutrients [6].

One of the most widely used techniques is image analysis using unmanned aerial vehicle (UAV) or satellite images [7,8,9,10]. NASA launched the first Earth satellite named Soil Moisture Active Passive (SMAP) satellite designated to monitor soil moisture across the globe [11]. Other methods include optical sensors which have a great sensitivity towards nitrogen sensing but greatly suffer from bulky spectrometer hardware and site-specific calibration and lack accuracy for detecting nutrients that are not fully observed in the Vis-NIR region [12,13]. Another approach uses a robotic platform that scans a specific field, detecting different vegetation and assessing irrigation cycles for the different fields [14].

These methods require collecting thousands of images stitched together and high processing power to analyze the data. On the other hand, miniaturized sensors require a microcontroller and a battery in a handheld device to measure the soil nutrients [15,16,17,18,19,20,21]. The simplicity of the hardware allows for weeks or months of data collection before the batteries need to be replaced. The sensors are calibrated on the soil samples, ensuring high measurement accuracy in situ. Screen-printed electrodes (SPEs) are the most common type of sensors used for their low cost, portability, and ease of insertion in the soil.

However, there has not been a long-term study on the lifetime or reliability of these sensors in situ. This work showcases the first-of-its-kind real-time continuous in situ soil nitrate measurement through the use of electrochemistry. This work presents an ion-selective electrode (ISE) incorporating tetradodecylammonium (TDDA) nitrate to increase selectivity towards nitrate ions [18,22,23]. Joly et al. have demonstrated a ChemFET device capable of measuring nitrate concentrations in the range of 10^−1.5^–10^−5.5^ M with a sensitivity of 56 mV/pNO_3_. However, the fabrication process of the ChemFET device is complex with eight steps with seven different photomasks that require precise alignment in the micrometer range and must be conducted inside a cleanroom [18]. Kim et al. achieved R^2^ > 0.99 for measuring soil nitrate in soil extract and not the soil without modification. Moreover, the bulkiness of the sensor and the requirement for a separate reference electrode eliminates the use of this sensor in the field [22]. Zhang et al. presented the same concept as Kim et al. with two sets of electrodes, a working electrode with an ion-selective membrane (ISM) and a reference electrode, to achieve higher sensitivity and a lower limit of detection. However, the measurements are taken from soil extract rather than the soil as is. Moreover, they addressed a fluctuation in their measured potential where it never reaches a steady state. With the instability of the system and the fact that the sensor only operates in soil extract, this sensor is not suitable for long-term soil nitrate monitoring [23]. Jiang et al. developed a nitrate ISE using tetra-n-octylammonium bromide (TOA-bromide) [24], while Fayose et al. used tetradodecylammonium chloride (TDACl) [25]. Although these sensors have high sensitivity and selectivity, they measure the nitrate concentration in soil extract rather than directly in the soil in the field without any modification to the soil. Baumbauer et al. have demonstrated a screen-printed electrode (SPE) for measuring soil nitrate with a low-cost sensor [26]. The presented sensor measures soil nitrate in situ; however, the batch-to-batch variations require calibration for each sensor prior to use. However, they reported that the sensors are unreliable in soils with medium to high concentrations of calcium, which is present in many soil types and fertilizers as well.

All previous studies have used open circuit potential (OCP) to measure the concentration of nitrate. Open circuit potential measures the equilibrium state of soil, which technically depicts bulk micro-environment, and cannot gauge dynamic soil phenomena. This study, on the other hand, utilizes electrochemical impedance spectroscopy (EIS) to gauge the soil dynamics, which is not only scientifically significant, but also relevant to building an Internet of Things (IoT)-enabled impedimetric platform for soil signal quantification. As soil textures are widely different, three calibrated dose–response curves have been built using three different soil textures. Singh et al. have demonstrated that different soil types behave differently under external stimulus [27]. Although they used square wave voltammetry (SWV) to measure pH in soil, the same concept applies to measuring other nutrients or ions in the soil. There, three different soil textures are chosen to cover the entire soil texture triangle; they are clay, sandy loam, and loamy clay, as indicated by the blue circles in Figure 1b. A 7-day study was conducted to showcase the capability of the sensor for continuous monitoring. All results were compared to the gold standard cadmium reduction method [28]. Finally, this work is the first proof of the feasibility of integrating electrochemical sensors in a monitoring system for real-time continuous tracking of the dynamic soil ecosystem.

## 2. Materials and Methods

### 2.1. Materials

Carbon screen-printed electrodes (Dropsens DRP 11L), with carbon working and counter electrodes and a Ag/AgCl reference electrode, were bought from Metrohm (Herisau, Switzerland). Tetradodecylammonium nitrate (TDDA), high-molecular-weight poly (vinyl chloride) (PVC), 2-nitrophenyl octyl ether (NPOE), tetrahydrofuran (THF) stabilized with BHT, potassium chloride, and sodium nitrate were bought from Sigma Aldrich (Burlington, MA, USA). An EmStat Pico module (portable potentiostat) was purchased from PalmSens (Houten, the Netherlands). Gaussian software Gaussian 16 W, version 1.1 (Wallingford, CT, USA), was used to perform the computational study. Illustrative sketches were drawn using BioRender. All statistical analysis was conducted using GraphPad Prism, version 9.3.1 (San Diego, CA, USA).

### 2.2. Soil Sample Preparation

Clay, loamy clay, and sandy loam soils were used to build the sensor calibration curve. These soil types provide full coverage of the soil texture triangle. The full flow is illustrated in Figure 2c. Air-dried soil was ground and filtered through a 2 mm mesh to acquire fine soil particles. Dilutions of sodium nitrate (NaNO_3_) dissolved in deionized (DI) water were prepared to acquire 9 samples with final nitrate concentrations of 0, 2, 8, 16, 32, 64, 128, 256, and 512 ppm. A mixture of 2 mL of soil and 1 mL of the corresponding NaNO_3_ solution was prepared for each soil type. The air-dried soil samples used had a nitrate concentration of less than 1 ppm, which is considered the baseline or 0 ppm sample. All soil samples were prepared the day before the experiment began to provide adequate time to ensure all samples were homogeneous.

### 2.3. Electrode Preparation

To prepare the nitrate ion-selective coating, 22.5 mg of PVC, 30 mg of NPOE, and 7.5 mg of TDDA were dissolved in 275 μL of THF, as illustrated in Figure 2a. The solution was mechanically stirred for 30 min followed by 20 min of sonication in a water bath at room temperature. These two steps were repeated until a homogeneous clear solution was obtained. Afterward, 2 μL of the solution was drop cast onto the working electrode, as shown in Figure 2b. The sensors were left to dry at room temperature for 4 h to ensure complete evaporation of the THF solution. The sensors were then stored in 0.01 M NaNO_3_ solution overnight before the experiments were started. After the experiment, the sensors were stored in fresh 0.01 M NaNO_3_ solution until the next experiment.

### 2.4. Hardware Development

The probe and hardware block diagram are shown in Figure 1c. The probe consists of a microcontroller (component 1), a potentiostat (component 2), an SD card module (component 3), a battery (component 4), and a sensor (component 5). The probe measures 46 cm in length and is split into three sections. Section i houses the electronics at the top 5 cm. Section ii allows the adjustment of the depth of the sensors from 10 cm down to 40 cm, while section iii houses the sensor. A Connfly cable connects the electronics at the top to the sensor location at the bottom of the probe. An Arduino MKR zero is used as a master to control the EmStat Pico and store the data on an SD card. The EmStat Pico derives its power through the MKR zero. The MKR zero was chosen as it comes equipped with an SD card module, thus negating the requirement of adding a separate SD card module and simplifying the programming complexity. The MKR zero has low operational and hibernation power consumption making it perfect for portable solutions. Once the battery is connected, the MKR zero runs a preloaded code that sends a script to the EmStat Pico. The Pico runs EIS and then returns the frequency, real impedance, and imaginary impedance data back to the MKR zero, and the data are saved on the SD card. The SD card module is only powered up when saving the data and immediately powered down to prolong the battery life. Once the measurement is complete, the MKR zero sends a script that places the Pico in hibernation. The MKR zero then hibernates and wakes up after the preset time has passed to repeat the process.

### 2.5. Experimental Design

A printed circuit board (PCB) was designed to host the EmStat Pico and provide connectivity to a computer through a USB cable with a connector to plug the sensor in as shown in Figure 1c. All measurements were conducted using the PSTrace software provided by PalmSens. The prepared soil samples were incubated on the sensors for 5 min before any measurement was taken. Electrochemical impedance spectroscopy (EIS) was run from 50 kHz to 5 Hz with an amplitude of 10 mV and 0 V DC bias. All plotting and statistical analyses were performed using the statistical and graphing software GraphPad Prism (Graph Pad Software Inc., La Jolla, CA, USA).

## 3. Results and Discussion

### 3.1. TDDA Gaussian Simulation for Visualization of Its Interaction

Soil is a reservoir of ions, and computational tools can be used to visualize the interaction between the ions. The ionophore, more technically, the selectophore, used in this study is the nitrate salt of the tri-dodecyl methyl ammonium (TDDA) ion, which is a zwitterionic species having the big cationic TDDA and anionic nitrate part.

The species also tends to interact with other ions majorly present in the soil, including ammonium, phosphate, potassium, and chloride. To understand the competitive interaction of TDDA–nitrate, we have optimized the TDDA-N by putting it in a hostile environment surrounded by other major ions present in the soil, and the Gaussian simulation has been performed using Hartree–Fock Method with basis set 6-31G-(d). The optimized structure of TDDA–nitrate with other ions is depicted in Figure 3a, and the simplified chemical depiction is demonstrated in Figure 3b. Figure 3c depicts the HOMO-LUMO orbital of the TDDA complex with all the competitive ions in its vicinity.

The result depicts that TDDA has a very strong affinity towards NO_3_^-^ having two strong noncovalent interactions with TDDA moiety. NH_4_^+^ has a strong affinity towards phosphate and can be seen interacting with phosphate and TDDA. Cl has a slight interaction with TDDA, but there is a negligible chance of the presence of Cl as Cl^-^ in soil, and hence, the formation of KCl is inevitable. Figure 3c depicts the HOMO-LUMO representation of the simulated complex, and it can be seen that the HOMO electron cloud is surrounded over H_2_PO_4_^−^, whereas LUMO is surrounded over TDDA–nitrate, which depicts suitable electron transfer from H_2_PO_4_^−^ to NO_3_^−^, which depicts strong interaction of TDDA with nitrate in the competitive micro-environment filled with other ions.

### 3.2. Sensor Calibration

The sensor response was calibrated against known doses prepared as prescribed previously to cover the range from 0 ppm to 512 ppm. As the PVC membrane contains TDDA, nitrate ions have the highest probability of binding to the ionophore. This change in charge on the electrode surface can be measured using EIS, as depicted in Figure 4a,b. As the nitrate concentration in soil increases, the impedance of the electrical double layer (EDL) decreases. The calibrated dose response (CDR) is calculated using Equation (1), where the percentage change in impedance, $\%∆Z_{mod}$, is between the sample with <1 ppm nitrate concentration, denoted as $Z_{0}$, and the measured impedance at every concentration, denoted as $Z_{m}$. Due to the slower diffusion rate in soil, every sample was incubated on the sensor for 5 min before measurement.
(1)$$\%∆Z_{mod}=\frac{100\times {(Z}_{m}-Z_{0})}{Z_{0}}$$

This ensures that the EDL reaches equilibrium. Measurements were taken in 1 min intervals up to 15 min, and the impedance change was negligible around the 4 min mark. Thus, 5 min was chosen as the incubation period. The mean and standard deviation were used in all reported results and plots. As shown in Figure 4c–e, the sensor’s sensitivity varies between the different soils, which is related to the density and porosity of the different soil textures. Clay soil had a lower standard deviation across sensors with low resolution, while the opposite was observed in sandy loam and loamy clay soils. All three calibration curves have an R^2^ > 0.958, indicating the high accuracy of the model in predicting the accurate concentration of nitrate in different soil types. The United States Department of Agriculture (USDA) reports healthy farmland soil levels year-round between 10 ppm and 70 ppm [29]. Therefore, the sensitivity of the sensor in this range is of high importance. Using linear regression on the data in the range of 8–64 ppm yields a sensitivity of 0.49% per ppm, 0.36% per ppm, and 0.41% per ppm for sandy loam, clay, and loamy clay soil, respectively. This means that a 5 ppm increase in concentration correlates to a decrease in impedance by ~900 Ω, ~650 Ω, and ~900 Ω for sandy loam, clay, and loamy clay soil, respectively. With the high accuracy of the potentiostat used illustrated by the low standard deviation plot in Figure 4c–e, these differences are significant and accurately measured.

### 3.3. Cross-Reactivity Experiment

Although the ionophore is specific to nitrate ions, as validated by computational results too, it is not immune to changes in the presence of other ions’ concentrations. For this purpose, a cross-reactivity study was set up. A high specificity eliminates incorrect reporting of nitrate concentration due to other ions interfering with the electrical double layer. Three different samples were prepared from the same sandy loam soil stock; one sample had <1 ppm nitrate (labeled as “0 ppm”), another sample had 16 ppm nitrate, and the last sample had 25 ppm of potassium with 25 ppm of phosphorus as well (labeled as “cocktail”). Potassium and phosphorus were chosen as they change frequently, similar to nitrate, while other ions such as carbon change slowly over months.

Figure 5 shows the percentage impedance change of the measured impedance for every sample in the order that they were added. The results show that the impedance stays within ±4% from the <1 ppm sample while having a −21.3% change for the 16 ppm sample. The −4% change from the baseline $Z_{0}$ translates to a nitrate concentration of 3 ppm which in all agriculture soils would be denoted as extremely low.

### 3.4. Validation

The efficacy of the system in providing an accurate nitrate concentration from the measured impedance was determined, and the results are shown in Table 1. The nitrate concentration of various soil samples of different textures was measured using the reference cadmium reduction method and compared to the measured nitrate concentration using the proposed sensor. The samples measured by the sensors had their water content adjusted to 15–50% of the sample’s weight, mimicking agriculture conditions. All measurements were performed using three different sensors (N = 3). Figure 6a shows Pearson correlation analysis between all the measured nitrate values using the proposed sensor system plotted on the *y*-axis and the reference nitrate values plotted on the *x*-axis, showing a Pearson r of 0.992. Figure 6b shows the two-way ANOVA between the reference nitrate and measured nitrate for the various soil textures.

All samples had a p-value of higher than 0.05 excluding one sample (*p*-value = 0.039), indicating an insignificant difference between the reference method and measured nitrate. However, reporting 9.8 ppm rather than the actual 11.9 ppm concentration is insignificant in real life but is statistically significant due to the small values being compared. These results show that creating three calibration curves of the textures at the three corners of the soil texture triangle to cover all soil textures is viable. The measured averages, standard deviation, and calculated error percentage are reported in Table 1. The proposed sensor had an error rate of less than 20% across all samples, indicating it is suitable for in situ measurement of soil nitrate.

### 3.5. Real-Time Continuous In Situ Soil Nitrate Monitoring

A temporal study over 7 days was set up. Figure 7a demonstrates the study setup. A 19 L bucket was used to mimic an agriculture field. The soil used was from a grazing site in Overton, Texas, US, with a nitrate concentration of ~4 ppm. The moisture was maintained at 15–60% throughout both studies, mimicking an irrigation cycle in the field.

The probe was inserted 20 cm under the soil surface so that the sensors were positioned at 15 cm, where most roots of small plants would be located. The electronics were housed at the top of the probe, as shown in Figure 7b. A measurement was recorded every hour for 7 days, as plotted in Figure 7c. It was observed that the sensor requires 12–24 h before stabilizing around the mean. Once a steady state is achieved, the measurements are within ± one standard deviation from the mean, indicating excellent reliability. Table 2 summarizes the mean, standard deviation, and coefficient of variation for all data points. A coefficient of variance of 17.6% indicates high stability in the measured nitrate. To validate the results, the measured values are compared to the cadmium reduction as the reference method. The error rate is less than 5%, showing excellent accuracy and reliability. This study also highlights the robustness of extracting nitrate concentration from EIS as compared to open circuit potential (OCP) which is commonly used with ISE sensors. The water content in the study was dynamically changing due to evaporation from the soil surface down to 15%. At this point, water was added to the soil, mimicking an irrigation cycle. Although this occurred multiple times throughout the 7-day study, the measured nitrate concentration stayed within one standard deviation from the mean. The other takeaway from this temporal study is the rigidity of the EIS method against degradation of the reference electrode that occurs as the sensor sits in the soil for many days.

## 4. Conclusions

An ion-selective screen-printed electrode is presented for in situ soil nitrate sensing. The proposed sensor does not require any sample pretreatment, thus accurately measuring the nitrate of unbuffered soil samples in the range from 8 ppm to 512 ppm. Electrochemical impedance spectroscopy (EIS) provides a nitrate-dependent response across the desired range irrespective of the soil type. More importantly, EIS provides immunity at frequencies higher than 200 Hz against environmental noise that is dominant at lower frequencies. Three calibration curves were built to cover all soil types in the soil texture triangle. The three calibration curves had an R^2^ > 0.957. Sensor performance was validated against the gold standard cadmium reduction and had an error rate of less than 20% across the different soil types, even those that were not used in building the calibration curves. The sensor sensitivity was optimized for the important range of 10–70 ppm where nitrate concentrations reside year-round in agricultural soil. Using linear regression on the data in the range of 8–64 ppm yields a sensitivity of 0.49% per ppm, 0.36% per ppm, and 0.41% per ppm for sandy loam, clay, and loamy clay soil, respectively. An interference study was conducted to showcase the selectivity of the sensor among other dynamically changing nutrients, namely potassium and phosphorus. A 7-day-long study also showed the reliability and stability of the sensor in measuring nitrate concentrations over long periods with an error rate of <5% and a coefficient of variance of <20%.

## Figures and Tables

**Figure 1 micromachines-14-01314-f001:**
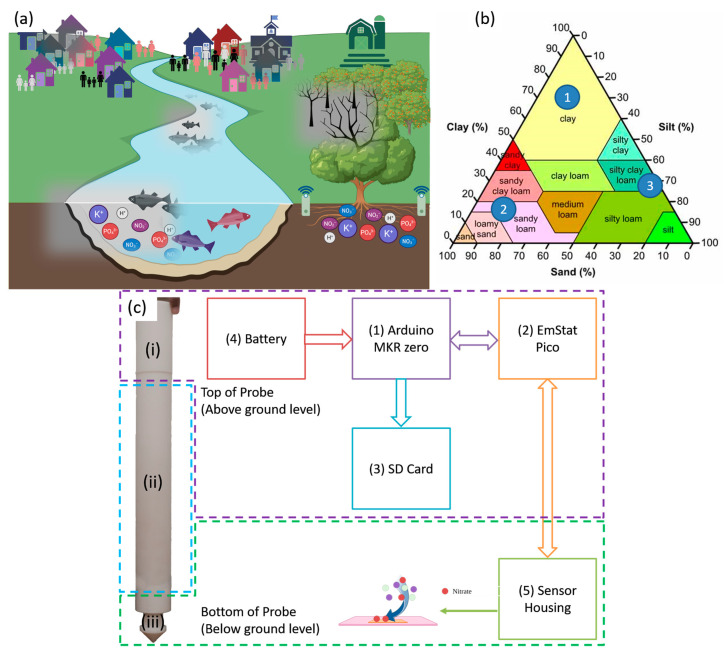
(**a**) Illustration of how lack of real-time monitoring of soil nitrate affects humanity; (**b**) the soil texture triangle provides a breakdown of the different soil types, where the numbers indicate the 3 chosen soil types for sensor calibration; (**c**) block diagram illustrating the hardware and probe used in this study.

**Figure 2 micromachines-14-01314-f002:**
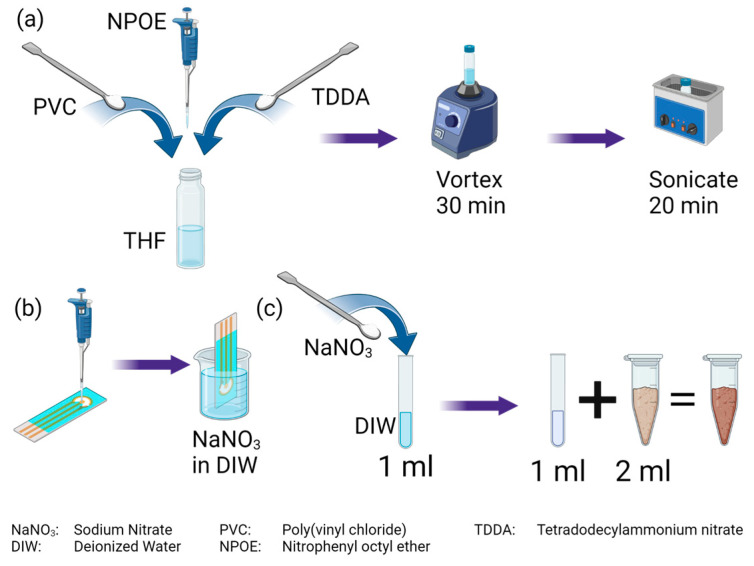
(**a**) The nitrate ion-selective coating was fabricated by mixing PVC, NPOE, and TDDA in THF then stirred and sonicated until a clear homogeneous solution was obtained. (**b**) Drop casting of the ion-selective coating onto the working electrode and then storage in NaNO3. (**c**) Soil sample preparation to obtain 9 different concentrations of nitrate in the different soil types.

**Figure 3 micromachines-14-01314-f003:**
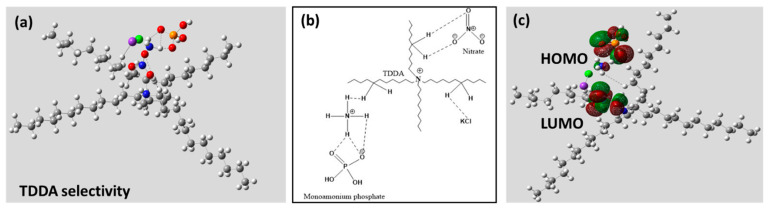
(**a**) Gaussian output of TDDA–nitrate complex with competitive ions depicting selective interactions of nitrate with TDDA, strengthens the proposed hypothesis. (**b**) Chemical architectural depiction to simplify the interactions. (**c**) HOMO-LUMO representation.

**Figure 4 micromachines-14-01314-f004:**
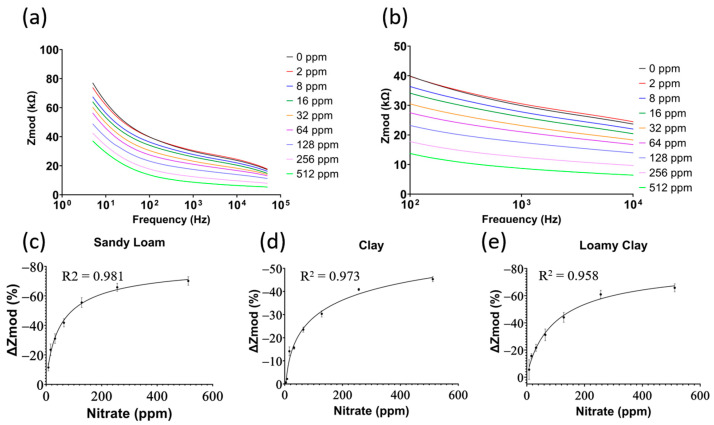
(**a**) Raw signals of electrochemical impedance spectroscopy for sandy loam soil; (**b**) zoomed-in view of the 1 kHz range; the calibrated dose response for (**c**) sandy loam soil, (**d**) clay soil, and (**e**) loamy clay soil (N = 3, mean ± standard deviation).

**Figure 5 micromachines-14-01314-f005:**
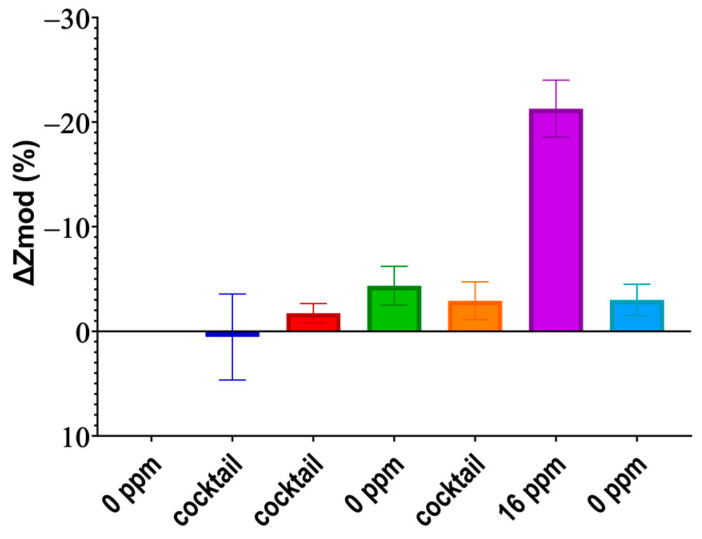
Cross-reactivity of phosphorus and potassium on the nitrate ISE plotted in the sequence of dosing.

**Figure 6 micromachines-14-01314-f006:**
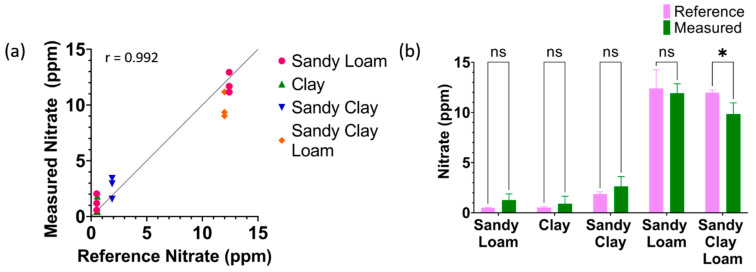
(**a**) Pearson’s correlation between measured nitrate using proposed sensor and cadmium reduction as the reference method; (**b**) *t*-test results between the reference method and measured values (N = 3, mean ± standard deviation, ns: *p* > 0.05, *: *p* < 0.05).

**Figure 7 micromachines-14-01314-f007:**
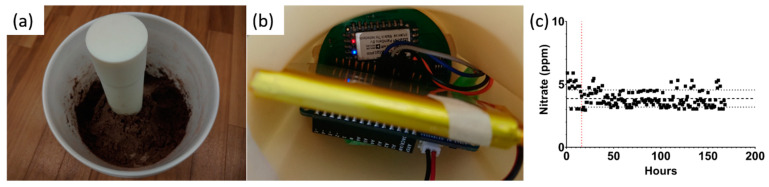
(**a**) Probe set up in the 19 L bucket, (**b**) the hardware housed at the top of the probe, and (**c**) measured nitrate over 7 days in sandy loam soil.

**Table 1 micromachines-14-01314-t001:** Sensor validation results.

**Sample #**	**Soil Type**	**Calibration Curve**	**Reference Nitrate (01.12.22)**	**SD**	**Measured Nitrate (06.09.22)**	**SD**	**Error (%)**
1S1	Sandy Loam	Sandy Loam	0.50	0.03	1.27	0.62	<4 ppm
1S2	Sandy Clay Loam	Clay	11.99	0.25	9.84	1.12	17.9
**Sample #**	**Soil Type**	**Calibration Curve**	**Reference Nitrate (03.19.22)**	**SD**	**Measured Nitrate (06.09.22)**	**SD**	**Error (%)**
2S1	Sandy Clay	Sandy Loam	1.88	0.21	2.64	0.97	<4 ppm
2S2	Clay	Clay	0.55	0.08	0.91	0.74	<4 ppm
2S3	Sandy Loam	Sandy Loam	12.41	1.85	11.92	0.94	3.95

**Table 2 micromachines-14-01314-t002:** Statistics and validation data for both temporal studies.

**Descriptive Statistics**	**Mock Study**	
Number of values	168	
Mean	3.87	
Std. deviation	0.682	
Std. error of mean	0.0527	
Coefficient of variation	17.6%	
**Validation**	**Measured**	**Reference**
Mean	3.87	~4
Std. deviation	0.682	-
Error rate	3.25%	

## Data Availability

Data can be shared only with the sponsor’s permission.

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
