# Peer review of "Electrochemical Soil Nitrate Sensor for In Situ Real-Time Monitoring"

_micromachines, 2023, doi:10.3390/mi14071314_

Round 1

Reviewer 1 Report

Eldeeb et al., have demonstrated a soil nitrate sensor. Sensor works based on a nitrate ion-selective membrane that is responsive to soil nitrate concentration.  Studies were conducted on three different soil textures including Clay, Sandy Loam, and Loamy 15 Clay. Authors claimed that the sensor based carbon electrode can detect soil nitrate for 7 days. I would recommend it for its publications after a major revision. Specific comments are given below

1.       Why did authors use a traditional screen printed carbon electrode to develop a soil sensor? Introduction is not summarized with the state of art technology for ISM-based soil nitrate sensors. Authors cite existing works such as a) https://iopscience.iop.org/article/10.1149/1945-7111/ab69fe/meta; b) https://acsess.onlinelibrary.wiley.com/doi/full/10.1002/saj2.20226; c) https://pubs.acs.org/doi/abs/10.1021/acsami.9b07120

2.       For in-situ long term measurement, how authors validate the quality of reference electrode stability.

Reviewer 2 Report

Title: Electrochemical soil Nitrate sensor for in-situ real-time monitoring

Mohammed Eldeeb and team reported that innovate technology for monitoring the in-site real time monitoring. They reported ion-selective electrode (ISE) electrochemical soil nitrate sensor that utilizes electro-chemical impedance spectroscopy (EIS) for direct real-time continuous soil nitrate measurement without any soil pre-treatment.

They found different results in different soil textures (Clay, Sandy Loam, and Loamy Clay).

Results validated with R2  > 0.97 for the various soil textures in the nitrate.

Abstract: excellent

Keywords: ok

Introduction: UNSDGs add some latest references L 25-40

Figure- 1 : outstanding

Pl add objective of study in the end of introduction section L78-79

Material & Methods: add one conceptual figure with including all steps of M & M

Add statically tools

Results & discussion: add quantify data in results section.

Discussion is missing add it with relevant references (major)

Conclusion: need quantify data

Overall, the manuscript is written well, need major modification is discussion and add quantify data in results, and conclusion section.

Overall, the manuscript is written well, need major modification is discussion and add quantify data in results, and conclusion section.

Round 2

Reviewer 1 Report

This revised version of the manuscript can be accepted for its publication.